# Physical and Biomechanical Relationships with Countermovement Jump Performance in Team Sports: Implications for Athletic Development and Injury Risk

**DOI:** 10.3390/sports13080277

**Published:** 2025-08-20

**Authors:** Moses K. Bygate-Smith, C. Martyn Beaven, Mark Drury

**Affiliations:** 1School of Sport and Human Movement, Te Huataki Waiora Division of Health, University of Waikato, Tauranga 3116, New Zealand; 2Recreation Centre, Lincoln University, Lincoln 7647, New Zealand; martyn.beaven@waikato.ac.nz; 3Faculty of Health, University of Canterbury, Christchurch 8041, New Zealand; mark.drury@canterbury.ac.nz

**Keywords:** countermovement jump, athletic performance, human characteristics, team sports, sports medicine

## Abstract

Background: Several physical qualities have been linked to countermovement jump (CMJ) performance. However, the relative importance of each of these factors is unclear. (1) Objectives: The present systematic review sought to evaluate the characteristics associated with CMJ performance in adult team-sport athletes. (2) Methods: A comprehensive search of three databases and the grey literature yielded 18 articles that met the inclusion criteria. Pearson’s correlation coefficient was used to assess statistically significant relationships and interpreted as negligible (0.00–0.10), weak (0.10–0.39), moderate (0.40–0.69), strong (0.70–0.89), and very strong (0.90–1.00). (3) Results: Eighteen articles remained eligible, with an average quality score of 76% ± 14 on the Joanna Briggs Institute critical appraisal index. The strongest correlations reported included time-to-bottom, time-to-peak force, knee extension peak power at 180 °/s, and squat jump height. (4) Conclusions: The conclusions drawn from this study suggest that, to maximize CMJ performance, priority should be given to movement biomechanics and lower-body power whilst considering individual braking-phase strategies. These findings may inform training programs aimed not only at enhancing athletic performance but also at reducing injury risks associated with poor jumping mechanics in team sports.

## 1. Introduction

Vertical jumping is a crucial skill that is prevalent across a wide range of team sports such as basketball, soccer, volleyball, and handball [1,2,3,4,5]. Even in team sports without vertical jumping demands within match play, vertical jumping variations such as the countermovement jump (CMJ) are often utilized for monitoring and enhancing power performance [6,7,8,9,10]. However, the CMJ not only has verified links with performance but also injury risk [11,12,13], such as orthopedic injuries like patellofemoral pain syndrome and osteoarthritis [12]. Therefore, it can be reasonably stated that understanding the physical underpinnings of maximizing CMJ height, speed, and other related metrics can be reviewed as being crucial for sports performance.

Several physical qualities have been linked to CMJ performance over the years. Specifically, features related to physical structure have been linked to CMJ performance, such as height [3,14,15], weight [8,14,16,17], and muscle architecture [18,19,20]. For example, Pocek and colleagues [15] found that female volleyball players that were taller and heavier also jumped significantly higher compared to shorter and lighter players. Evidence has also shown relationships between CMJ performance and various strength [3,21,22,23], power [3,24,25,26], change of direction (COD) [27,28], and sprint qualities [29,30,31,32]. Sheppard et al. [3] identified a strong linear correlation between relative power clean one-repetition maximum (1 RM), CMJ relative maximal strength, and CMJ height, showing that larger relative values are associated with greater jump heights. Suarez-Arrones and colleagues [28] saw that greater CMJ heights were correlated with faster 10-metre linear sprint, 180° COD, and 90° COD test times. From a biomechanical perspective, previous research has also reported that propulsive-phase power and individual joint-extension power are associated with enhanced CMJ performance [33,34,35,36,37]. Specifically, Vanezis and Lees [10] found that higher jumpers were able to produce greater propulsive power relative to their body mass, particularly from their ankle and knee joints. The propulsive phase has previously been referred to as the concentric phase and can be defined as the upward portion of movement that occurs following the braking phase, prior to the jump take-off [38]. While multiple studies have confirmed links between various characteristics and CMJ performance, it is unclear what the relative importance of each of these factors is, particularly due to conflicting findings between authors. For example, although Pocek et al. [15] found that taller individuals jumped higher than shorter individuals, Sheppard et al. [3] found the opposite to be true. In terms of biomechanical factors, large heterogeneity has also been reported for certain metrics with examples such as braking impulse and knee flexion correlations with CMJ height ranging between negligible and very strong in their magnitudes [38,39,40,41,42]. These findings make it difficult to ascertain the overall importance of certain characteristics for maximizing CMJ performance and therefore which aspects to prioritize for scouting, training, or rehabilitation. Furthermore, to our knowledge, no systematic review has been conducted looking at a broad spectrum of characteristics and their relationships with CMJ performance.

Therefore, the aim of this systematic review is to evaluate the association between various characteristics and CMJ performance in adult team-sport athletes while also understanding the relative importance of each characteristic for maximizing CMJ performance. The different types of CMJ performances assessed include jump height, time-to-take-off, reactive strength index, rate of force development, and other related metrics. Based on the initial review of the literature, characteristics are classified based on being anthropometric, morphological, strength, power, COD, sprint, global CMJ, and CMJ joint-related features.

## 2. Materials and Methods

### 2.1. Study Design

This systematic review was conducted according to the PRIMSA statement guidelines (Preferred Reporting Items for Systematic Review and Meta-Analyses) [43]. The completed PRISMA checklist is provided in the Appendix A [44]. Ethical approval was granted from the institutional review board on 29 July 2024 (HREC2024#24). The review was not registered.

### 2.2. Search Strategy

Three electronic databases (PubMed, Scopus, and SPORTDiscus) were searched systematically using the following keywords and Boolean operators: “vertical jump OR countermovement jump AND performance* OR abilit* OR biomechanical analysis OR strategies OR characteristics OR factors OR differences OR measures OR predict* OR anthropometr* OR morpholog* OR body constitution.” A grey literature search was also conducted using the University of Waikato Library database and the Google search engine to find the supporting literature. All sources were last searched on 10 May 2024.

### 2.3. Eligibility Criteria

Studies were included if they (1) were original peer-reviewed cross-sectional articles written in English; (2) published in the last 15 years; (3) reported on characteristics in relation to a CMJ analysis; and (4) involved healthy adult (>19 years) team-sport athletes of either gender. Studies were excluded if they (1) involved injured or physically impaired participants; (2) the mean sample age of participants was older than 44 years old or younger than 19 years old; (3) most of the subjects (>50%) were from an individual-sport or recreational-level background; or (4) focused on drop jump, squat jump, approach jump, broad jump, or other jumping variations that were not the CMJ.

### 2.4. Study Selection

One author (M.K.B.S.) conducted the study screening and selection process by identifying articles during the database search and removing duplicates that were identified. Following duplicate article removal, the remaining studies were screened by their titles and abstracts and then finally received full-text screening based on the review eligibility criteria. The study selection process was replicated for articles that were gathered through the grey literature search (Figure 1). If article inclusion uncertainty was encountered, an additional investigator (C.M.B.) was consulted for a secondary opinion.

### 2.5. Assessment of Reporting Quality

The methodological quality of each article that met the eligibility criteria was assessed by one author (M.K.B.S) using the Joanna Briggs Institute (JBI) checklist for analytical cross-sectional studies [45]. The checklist has been previously used to assess the quality of articles included in systematic reviews [46] and provides an overall score out of 100% based on an eight-question criterion concerned with bias in design, conduct, and analysis. For each criterion, either a yes, no, unclear, or not applicable answer was awarded, and articles with a higher percentage of yes answers were indicated to be of greater methodological quality. The quality scores of the studies were established based on a previous systematic review [47], where studies with a JBI score higher than 70% were rated as high-quality, between 50% and 70% were rated as moderate-quality, and scores lower than 50% were rated as low-quality. Articles that scored lower than 50% were excluded from the review.

### 2.6. Data Extraction

One author extracted the data (M.K.B.S.) using Microsoft Excel 2024 (Microsoft Corporation, Redmont, WA, USA). Participant characteristics such as sample size, gender sample distribution, mean age, mean height, mean weight, sport background, and competition level were extracted from each study. Intervention characteristics were also gathered, which included the testing equipment used, number of CMJ repetitions allocated, rest periods prescribed between attempts, jumping intensities, and verbal instructions delivered. Lastly, outcome measures were extracted and categorized as either being anthropometric, morphological, strength, power, COD, sprint, global CMJ, or joint-related CMJ characteristic.

### 2.7. Categorization and Presentation of Findings

CMJ performance characteristics were categorized as anthropometric, morphological, strength, power, COD, sprint, global CMJ, and joint-related CMJ. Anthropometric and morphological characteristics include features related to aspects of body structure. Strength characteristics include variables related to general force expression or gym-based movements with external load. Power characteristics refer to the general expressions of power or velocity features, such as other jumping variations or ballistic activities. COD and sprint characteristics include tests that directly assess linear acceleration, speed, or COD times, as well as other metrics derived from such tests. Global CMJ characteristics include kinetic and temporal measures generated by the whole system throughout all the CMJ phases. This includes the unweighing, braking, propulsive, and take-off phases (Figure 2). Finally, joint-related CMJ characteristics include specific joint outputs or displacements occurring from each of the CMJ phases.

### 2.8. Statistical Analyses

Pearson’s product-moment correlation coefficient was used with an alpha level set at ρ ≤ 0.05 to assess the characteristics where statistically significant associations with CMJ performance were observed. Additionally, the magnitude and direction of these relationships (r-value) were interpreted using the following effect size scale: negligible (0.00–0.10), weak (0.10–0.39), moderate (0.40–0.69), strong (0.70–0.89), and very strong (0.90–1.00) [48]. Ninety-five percent confidence intervals were generated to assess the certainty of the evidence presented.

## 3. Results

### 3.1. Selection of Studies

A total of 660 records were gathered from the selected databases, with an additional 6 articles being gathered from a grey literature search. Of the 666 total studies gathered, 117 duplicates were identified and removed, a further 507 articles were excluded during the title and abstract screening phase, and another 18 studies were removed following the full-text screening phase. Four of the six articles from the grey literature search were also excluded under the same criteria, leaving a total of 18 eligible articles for the review (Figure 1).

### 3.2. Assessment of Reporting Quality

Most articles (13/18) gained high (70%+) scores, which suggests high reporting quality according to the JBI checklist. Confounding influences were neglected or poorly considered across studies, with only three articles (15%) acknowledging and accounting for this. However, all other categories, such as having clearly defined participant inclusion criteria, detailed participant description, validity and reliable procedures, objective and standardized condition measurements, and appropriate statistical model application, scored well over the 70% threshold across studies. A summary of the study reporting quality scores can be seen in Table 1.

### 3.3. Participant Characteristics

A total of 3329 athletes were considered across the 18 studies, consisting of 1834 males, 1012 females, and 483 unspecified participants. Mean height, weight, and age, as well as playing level, are summarized in Table 2. Athlete sport backgrounds included American football (*n* = 619), basketball (*n* = 479), soccer (*n* = 462), baseball (*n* = 243), softball (*n* = 198), lacrosse (*n* = 162), volleyball (*n* = 136), tennis (*n* = 98), rowing (*n* = 97), rugby union (*n* = 30), hockey (*n* = 25), handball (*n* = 15), 3 × 3 basketball (*n* = 15), Australian-rules football (*n* = 14), and cheerleading (*n* = 9). Track and field (*n* = 329), swimming and diving (*n* = 163), golf (*n* = 47), cross-country (*n* = 49), fencing (*n* = 90), and wrestling (*n* = 24), although being individual sports, were included in the sample due to making up less than 50% of the demographic total, as specified in the eligibility criteria. The sports backgrounds of the remaining 25 athletes were not specified.

### 3.4. Intervention Characteristics

A wide range of equipment and protocols were used for CMJ analysis across studies. Specifically, thirteen articles used force platforms, five used marker-based motion capture, three used a yardstick jumping device, two used a jump mat, and one used an optical measurement device. Prescribed jump attempts ranged between 1–7 repetitions with rest periods ranging between 15–400 s. Most studies provided verbal instructions to participants prior to their CMJ attempts. A summary of these findings can be seen in Table 3.

### 3.5. CMJ Performance Characteristics

#### 3.5.1. Anthropometric and Morphological Characteristics

Eighteen total significant correlations between anthropometric and morphological characteristics and CMJ performance were reported in four out of the eighteen studies. Of the 18 total correlations, there were 15 different outcome measures, which included height, weight, body widths, body girths, muscle index, muscle-to-bone ratio, and muscle cross-sectional area of the mid-thigh. Although not being an anthropometric or morphological characteristic, sex and sex multiplied by weight were also reported as having a significant correlation with CMJ performance, with males and lower body weights being favorable [14]. Correlation coefficient values ranged between weak and strong. A summary of the findings can be seen in Figure 3.

#### 3.5.2. Strength Characteristics

Six significant correlations between strength characteristics and CMJ performance were reported in four of the eighteen studies. The outcome measures of the studies collectively focused on back squat measures, squat jump measures, and deadlift measures, with correlation coefficient values ranging between moderate and strong. A summary of the findings can be seen in Figure 4.

#### 3.5.3. Power Characteristics

Twenty-one total significant correlations between power characteristics and CMJ performance were reported in five of the eighteen studies. Of the 21 total correlations identified, there were 17 different outcome measures which focused on knee-joint isokinetic outputs, squat jump measures, cycling outputs, deadlift measures, push press, and medicine ball throws. Knee extension average power at 60 and 180°/s were significant correlations reported amongst the knee-joint isokinetic outputs, but r-values were not provided [56]. For all other variables, correlation coefficients ranged between moderate and very strong. A summary of the findings can be seen in Figure 5.

#### 3.5.4. COD and Sprint Characteristics

Thirty-nine total significant correlations between COD and sprint characteristics and CMJ performance were reported in seven of the eighteen studies. Of the 39 total correlations found, there were 13 different outcome measures, which consisted of a variety of different COD and sprint tests, as well as COD deficit, running power, and running force. Amongst the correlations identified, COD and acceleration test time and COD deficit at various distance splits were indicated; however, ρ-values or r-values were not provided [51]. For all other variables, correlation coefficients ranged between moderate and strong. A summary of the findings can be seen in Figure 6.

#### 3.5.5. Global CMJ Characteristics

Sixty-eight total significant correlations between global CMJ metrics and CMJ performance were reported in seven out of the eighteen studies. Of the 68 total correlations, there were 28 different outcome measures, which included a wide range of variables across the unweighing, braking, amortization, propulsive, and take-off phases of the CMJ. This included rate of force development, force, time, impulse, displacement, modified reactive strength index, power, acceleration, and force–time profile characteristics. Johnston et al. [40] and Sole et al. [42] collectively identified the braking to propulsive phase shape factor ratio, unweighing peak force, maximum vertical ground reaction force, and relative propulsive impulse as having significant correlations with CMJ performance among the total 68 variables identified, but they failed to provide r-values. For all other variables, correlation coefficients ranged between negligible and very strong. A summary of the findings can be seen in Figure 7.

#### 3.5.6. CMJ Joint-Related Characteristics

Forty-four total significant correlations between CMJ joint-related metrics and CMJ performance were reported in four of the eighteen studies included. Of the 44 total correlations, there were 30 different outcome measures that were reported, which included a wide range of specific lower-limb joint measures across the braking, propulsive, and take-off phases of the jump. Examples included time, net moments, acceleration, angles, power, work, velocity, and movement phase composition. Ankle dorsiflexion velocity, knee flexion velocity, power absorbed by the knee, ankle plantarflexion power, and the percentage of ankle power generation occurrence were among the correlations reported but were not provided with r-values [40]. For all other variables, correlation coefficients ranged between weak and strong. A summary of the findings can be seen in Figure 8.

## 4. Discussion

The aim of this systematic review is to evaluate the association between various characteristics and CMJ performance in adult team-sport athletes while also understanding the relative importance of each characteristic for maximizing CMJ performance. The aim of the systematic review was to assess the literature that exists regarding the relationship various characteristics have with CMJ performance in adult team-sport athletes while also understanding the relative importance of each characteristic presented. A total of 196 significant correlations of varying magnitudes were identified from the 18 eligible articles reviewed. Specifically, the characteristics that were associated with CMJ performance were 18 anthropometric and morphological, 6 strength-related, 21 power-related, 39 COD and sprint-related, 68 global CMJ-related, and 44 CMJ joint-related characteristics.

### 4.1. Anthropometric and Morphological Characteristics

Waist, thigh, and hip girth were reported as having the strongest correlations with CMJ performance out of all the anthropometric and morphological characteristics, demonstrating moderate to strong negative correlations. These findings do not come as a surprise given that maximum CMJ height is largely determined by greater take-off velocities generated by high impulses relative to bodyweight [38,39,42]. Thus, if there is additional mass to overcome, possibly indicated by larger waist, thigh, and hip girths, a greater impulse is required to jump high. Previous research has also noted the commonality of adipose tissue storage in these specific areas of the body [57]; therefore, this may explain why these factors presented greater negative relationships with CMJ performance. This is also reinforced by the negative relationship identified between weight and CMJ performance. Given the well-known negative association between excess weight, in particular fat mass, and markers of obesity such as cardiovascular diseases and metabolic diseases [57], it would be highly recommended that individuals looking to maximize their CMJ performance and overall markers of health should look to minimize any unnecessary body weight, especially if carrying higher percentages of body fat.

Muscle-to-bone ratio, muscle cross-sectional area of the mid-thigh, and other muscle-related markers closely followed behind waist, thigh, and hip girth in terms of the strength of their association with CMJ performance. The findings of a systematic review and meta-analysis which looked at the relationship between skeletal muscle architecture of the lower limbs and jumping ability noted that only vastus lateralis muscle thickness had a significant positive association with CMJ height [58]. However, as their review acknowledged, there was insufficient data to conduct analysis on other muscle groups and their architectural elements. Of note is the fact that findings from a study that failed to meet the eligibility criteria of this review due to having an excluded subject demographic (non-team-sport athletes) had conflicting results. Specifically, Earp et al. [19] found that the cross-sectional area of the vastus lateralis muscle was unable to predict jumping performances (*p* ≥ 0.05) while the lateral gastrocnemius could, although only to a trivial extent (*R*^2^ = 0.187). Considering the negative implications associated with reductions in muscle development, the link between the CMJ and various muscular features may require further exploring. If a strong and definitive relationship were present, this could provide an effective tool for public health screening on aging individuals who encounter reductions in performance due to sarcopenia [59]. Furthermore, as other connective tissue elements involved in force transmission have not been acknowledged from the review, it may be worthwhile for future investigations to focus on the contribution of these characteristics to CMJ performance as well. An interesting outcome from the review was the positive correlation between height and CMJ performance. These findings contrast to those of Sheppard et al. [3], who found that taller individuals performed worse than shorter individuals when it came to both the CMJ and the spike jump. However, given the mixed-participant sample of the review, it is possible that the best jumpers were made up mostly of males who are generally found to be taller and perform better than their female counterparts [1,27,42,45]. Future research should consider sex-specific relationships and involve a closer balance of males and females. Before developing conclusions on the role of anthropometric and morphological characteristics in CMJ performance, what must be considered is that only four studies out of the total reported on these types of features. Furthermore, the available data focused largely on anthropometric characteristics such as height and weight and had limited insights on body composition or morphological characteristics of the muscles and tendons.

### 4.2. Strength, Power, COD, and Sprint Characteristics

Many strength, power, COD, and sprint qualities were found to be associated with CMJ performance, such as relative back squat one-repetition maximum strength (1 RM), knee extension peak power at 180 °/s (KEPP180), and COD deficit demonstrating strong positive relationships. The strong link between strength, power, COD, sprint ability, and CMJ performance has been well documented in previous studies, with other investigations showing similar relationships [3,14,21,22,27,28,33,60,61]. These associations can be partially attributed to shared physiological mechanisms; for example, both the CMJ and COD ability involve a rapid application of braking impulse during the downward phase, followed by a fast reversal of the center of mass into propulsion. Given the link between sarcopenia and reductions in muscular performance [59], which occurs with aging, these findings suggest further implication for the use of the CMJ in public health screening settings.

Of the strength, power, COD, and sprint characteristics reported, KEPP180 had the strongest relationship with CMJ performance at a near-perfect correlation, followed by squat jump height. It is relatively well understood that the knee extensors are a significant contributor to jump height [12,41,49]. Furthermore, KEPP180 having a stronger association with CMJ performance than knee extension peak power at 60 °/s indicates that there is also a positive trend between the rate at which peak power is reached during knee extension and jump height. Of note is the fact that the relative back squat 1 RM and relative deadlift 1 RM not only had the strongest relationships with CMJ performance out of all the strength measures reported but also had identical r-values. Schiemann et al. [62], although not reporting on the back squat, found that the magnitude of the correlation between deadlift 1 RM and CMJ performance depended upon the sporting background of the athlete, with basketball players showing larger effect sizes compared to individuals from other sporting backgrounds. As basketball is a sport that consists of a high volume of jumping, perhaps these demands led to optimizing the skill of jumping to a greater extent than the other team-sport athletes and therefore facilitated better transfer of the strength derived from the deadlift. If these patterns exist for the deadlift, it is also possible that these trends may exist for other movements such as the back squat as well. Although being an old review, Baker [63] suggested that if the skill of jumping was optimized and a concomitant increase in strength was to follow, a significant improvement in jump height would occur. Alternatively, if the jumping skill was not optimized but the same increase in strength were to occur, then a decrement in jump height would unfold. Therefore, CMJ technical proficiency may determine the magnitude of the relationship with strength exercises.

### 4.3. Global CMJ Characteristics

Of the global CMJ metrics, time-to-bottom had the strongest correlation with CMJ performance, followed by time-to-peak-force. These observations suggest that the rate at which the unweighing and braking phases are performed, as well as how quickly a team-sport athlete can generate force, may be a key aspect to maximizing CMJ height. However, conclusions cannot be made with confidence due to downward-phase variables being underreported compared to propulsive-phase measures. Specifically, braking-phase variables were only reported 35 times versus 53 times for propulsive-phase metrics. Furthermore, a handful of studies from this review revealed that significant differences in CMJ braking-phase strategies were used, yet trivial-to-insignificant differences in jump height occurred between these groups [38,41,49]. For example, Rauch and colleagues [64] discovered three different CMJ kinematic braking-phase strategies in a large cohort of National Basketball Association (NBA) players and found no significant differences in jump height. These findings reinforce a further need for analysis to be conducted on the role that braking-phase variables play in maximizing CMJ performance, minimizing injury risk, and characterizing individual movement strategies that may exist.

### 4.4. CMJ Joint-Related Characteristics

Finally, the CMJ joint-related metrics that reported the strongest correlations with CMJ performance were the percentage of the movement in which peak dorsiflexion occurs (%PkDORSI), followed by the percentage of the time through the movement between peak hip and knee flexion (PHKT). Rauch and colleagues [41] found that greater %PkDORSI and PHKT led to a greater likelihood that basketball players would jump higher during the CMJ. However, this was specific to certain clusters of individuals within their sample and not the entire group. These findings further support the existence of alternative movement strategies during the braking phase of the CMJ and potentially calls for an individualized training approach to be considered. However, it appears that propulsive-phase strategies appear relatively consistent.

For example, the relative time delay between joint extensions, referred to by Chiu and colleagues [12] as a proximal-to-distal joint extension strategy with longer time delays between each lower-limb joint extension, had a strong correlation with CMJ height and was reported with relatively narrow confidence intervals. These findings agree with Cefai et al. [65], suggesting that to maximize CMJ height, there is a specific set of kinematic principles during the propulsive phase that must be met, whereas the braking phase affords greater degrees of freedom. A proximal-to-distal joint extension strategy may also have significant implications for injury risk mitigation, as Chiu and colleagues [12] mentioned that employing such a technique may minimize antagonist co-contraction and compressive forces at the knee joint, which could therefore reduce orthopedic injuries like patellofemoral pain syndrome or osteoarthritis.

Peak force during the braking and propulsive phases differed between each other in terms of their correlations to CMJ performance as well, with propulsive-phase peak force having a stronger relationship. These findings mimic those of González-Badillo and Marques [33], who found that propulsive force was higher and had a stronger correlation to CMJ height compared to braking force. Even though statistical significance was met for braking force application in their study, effect size values were highly heterogenous across all attempts. The authors suggested that this heterogeneity likely indicated varying degrees of importance for producing high braking forces to maximize CMJ height between individuals. As a low-magnitude relationship between braking force and jump height was indicated overall, this is in partial agreement with the current review’s findings.

### 4.5. Limitations

To the authors knowledge, this is the first systematic review which has comprehensively assessed a broad spectrum of characteristics associated with CMJ performance in adult team-sport athletes. However, while multiple relationships were identified, this review is not without limitations. Firstly, there was an imbalanced report on gender, with females only making up 30% of the sample. Furthermore, recreational-level and individual-sport athletes were primarily excluded from the review. This makes the findings of this review difficult to generalize across subject demographics. There was also an insufficient amount of data to conduct a meta-analysis. Therefore, caution should be taken prior to the interpretation of the review findings.

### 4.6. Practical Applications

The current synthesis of physiological and biomechanical characteristics that relate to enhanced CMJ performance can not only assist coaches, team-sport athletes, and medical staff working at the professional and elite levels but also cater to those operating within amateur, recreational, and general population environments. Through providing a framework on which factors to prioritize, this review can serve as a guide for enhancing performance and potentially minimizing injury risk. This includes recommendations for achieving an optimal balance of rapid movement times and force generation while using a proximal-to-distal joint extension strategy. However, caution should be taken when considering factors relating to the braking phase of movement or physical structure, as large heterogeneity and underreporting of these measures occurred. Furthermore, caution should be taken when generally applying the findings to different demographic groups, as outcomes may differ by sex and age and could potentially put such groups at risk. Overall, the CMJ can be used as a reliable, safe, and effective tool to assist in the development and monitoring of athletic performance, as well as the screening of obesity and age-related declines in performance.

## 5. Conclusions

The aim of the systematic review was to evaluate the association between various characteristics and CMJ performance in adult team-sport athletes while also understanding the relative importance of each characteristic presented. The findings of this systematic review provide valuable insights to coaches and strength and conditioning practitioners on the anthropometric, morphological, strength, power, COD, sprint, and biomechanical characteristics that relate to enhanced CMJ performance. Based on the results, adult team-sport athletes looking to maximize their CMJ performance should consider prioritizing their CMJ biomechanics, specifically by decreasing time spent during the unweighing and braking phases, as well as reaching peak force. Secondarily, addressing underlying qualities for enhancing lower-body power such as by increasing knee extension outputs at high angular velocities and squat jump heights will likely enhance CMJ-performance-related outcomes. However, the unique braking-phase movement strategy of an individual may also need to be considered when training to optimize jump performance, as large heterogeneity was identified for these types of metrics within the review. These specific findings suggest that an individualized approach should be taken when providing cueing and exercise interventions for enhancing the braking phase of movement to positively influence CMJ performance. However, future research should consider conducting further analysis on which braking-phase measures are important for enhancing jump performance, alongside further investigations into muscle- and tendon-related features, given how underreported they were. The CMJ can aid in the assessment of athletic performance, alongside screening in the public health sector.

## Figures and Tables

**Figure 1 sports-13-00277-f001:**
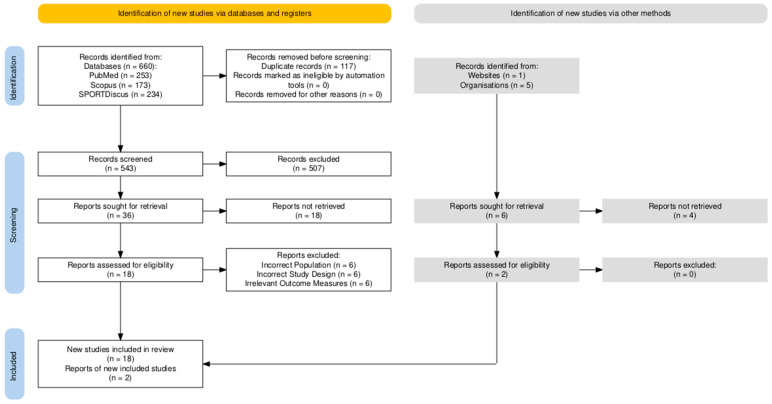
PRISMA flow diagram of the search strategy.

**Figure 2 sports-13-00277-f002:**
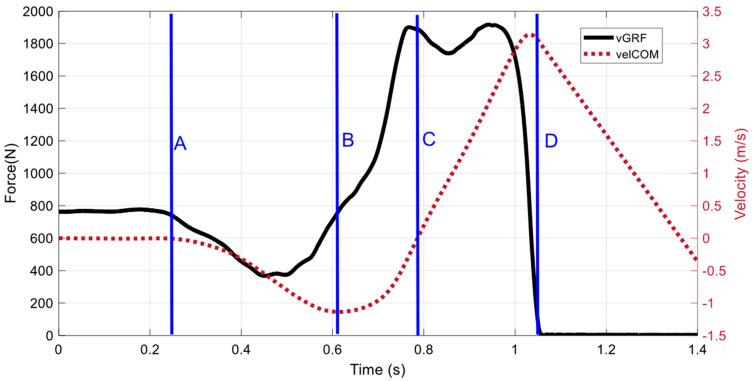
Countermovement jump (CMJ) phases [38]. (A) Unweighing phase, (B) braking phase, (C) propulsive phase, (D) take-off phase.

**Figure 3 sports-13-00277-f003:**
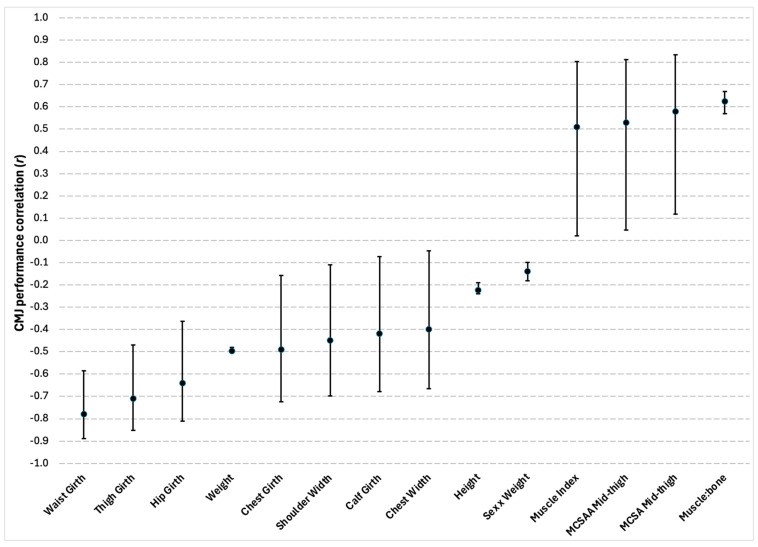
Correlations between the countermovement jump (CMJ) and anthropometric and morphological characteristics. Circles represent the coefficients of correlation (r), and the bars represent 95% confidence limits. All correlations were significant at *p* ≤ 0.05. MCSAA = muscle cross-sectional area adjusted; MCSA = muscle cross-sectional area; Muscle:bone = muscle to bone ratio.

**Figure 4 sports-13-00277-f004:**
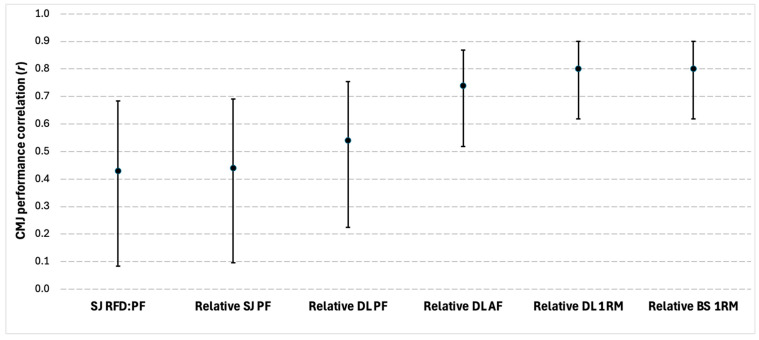
Correlations between the countermovement jump (CMJ) and strength characteristics. Circles represent the coefficients of correlation (r), and the bars represent 95% confidence limits. All correlations were significant at *p* ≤ 0.05. SJ RFD:PF = Squat jump rate of force development to peak force ratio; Relative SJ PF = relative squat jump peak force; Relative DL PF = relative deadlift peak force; Relative DL AF = relative deadlift average force; Relative DL 1 RM = relative deadlift one-repetition maximum; Relative BS 1 RM = relative back squat one-repetition-maximum.

**Figure 5 sports-13-00277-f005:**
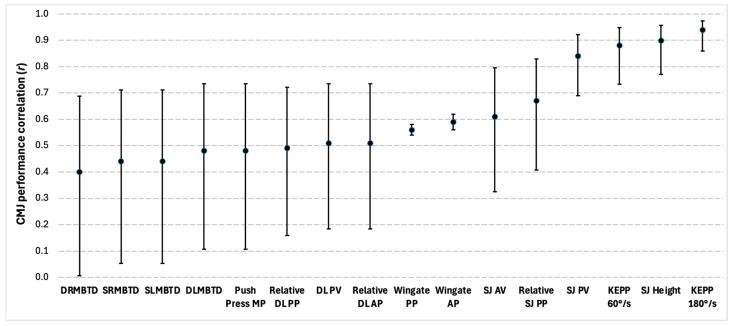
Correlations between the countermovement jump (CMJ) and power characteristics. Circles represent the coefficients of correlation (r), and the bars represent 95% confidence limits. All correlations were significant at *p* ≤ 0.05. DRMBTD = dynamic right medicine ball throw distance; SRMBTD = static reverse medicine ball throw distance; SLMBTD = static left medicine ball throw distance; DLMBTD = dynamic left medicine ball throw distance; Push Press MP = push press mean power; Relative DL PP = relative deadlift peak power; DL PV = deadlift peak velocity; Relative DL AP = relative deadlift average power; Wingate PP = wingate peak power; Wingate AP = wingate average power; SJ AV = squat jump average velocity; Relative SJ PP = relative squat jump peak power; SJ PV = squat jump peak velocity; KEPP 60°/s = knee extension peak power at 60 degrees per second; SJ Height = squat jump height; KEPP 180°/s = knee extension peak power at 180 degrees per second.

**Figure 6 sports-13-00277-f006:**
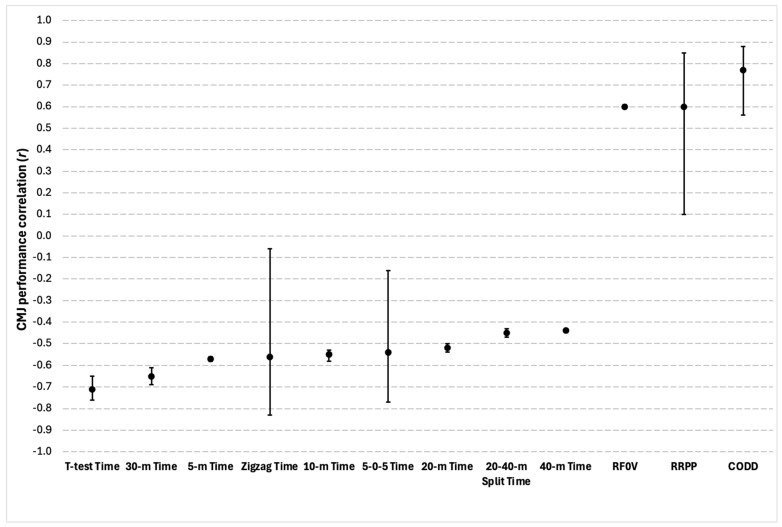
Correlations between the countermovement jump (CMJ) and COD and sprint characteristics. Circles represent the coefficients of correlation (r), and the bars represent 95% confidence limits. All correlations were significant at *p* ≤ 0.05. RF0V = relative running force at zero velocity; RRPP = relative running peak power; CODD = change of direction deficit.

**Figure 7 sports-13-00277-f007:**
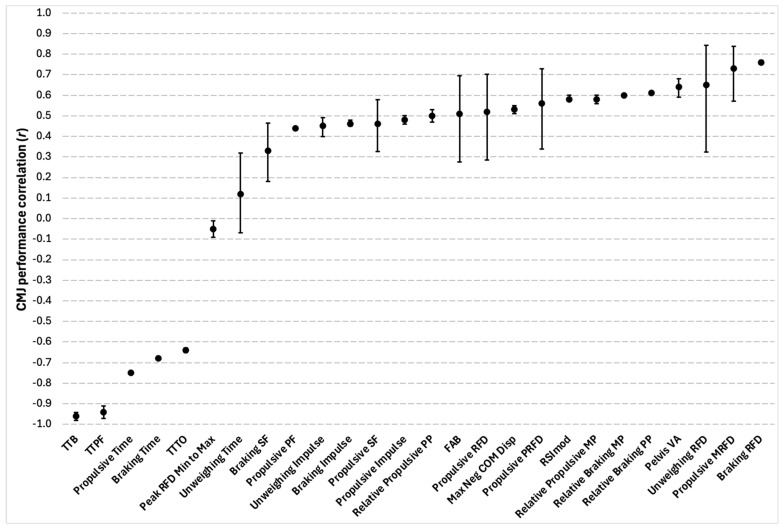
Correlations between the countermovement jump (CMJ) and global characteristics. Circles represent the coefficients of correlation (r), and the bars represent 95% confidence limits. All correlations were significant at *p* ≤ 0.05. TTB = time-to-bottom; TTPF = time-to-peak force; TTTO = time-to-take-off; Peak RFD Min to Max = peak propulsive rate of force development minimum force to maximum force; Braking SF = braking shape factor; Propulsive SF = propulsive shape factor; Relative Propulsive PP = relative propulsive peak power; FAB = force at bottom of countermovement; Max Neg COM Disp = maximum negative center of mass displacement; Propulsive PRFD = propulsive peak rate of force development; RSImod = modified reactive strength index; Relative Propulsive MP = relative propulsive mean power; Pelvis VA = pelvis vertical acceleration.

**Figure 8 sports-13-00277-f008:**
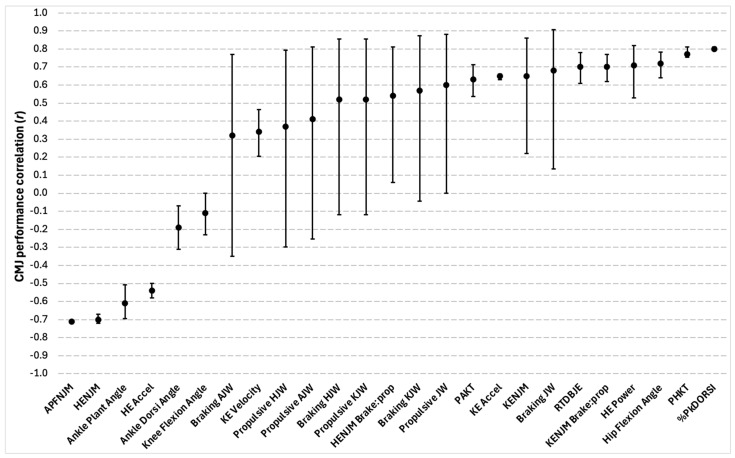
Correlations between the countermovement jump (CMJ) and joint-related characteristics. Circles represent the coefficients of correlation (r), and the bars represent 95% confidence limits. All correlations were significant at *p* ≤ 0.05. APFNJM = ankle plantarflexion net joint moment; HENJM = hip extension net joint moment; Ankle Plant Angle = ankle plantarflexion angle; Ankle Dorsi Angle = ankle dorsiflexion angle; Braking AJW = braking ankle joint work; KE Velocity = knee extension velocity; Propulsive HJW = propulsive hip joint work; Propulsive KJW = propulsive knee joint work; HENJM Brake:prop = hip extension net joint moment braking to propulsive phase ratio; PAKT = percentage difference of time through the movement between peak ankle dorsiflexion and peak knee flexion; KE Accel = knee extension acceleration; RTDBJE = relative time delay between joint extensions; PHKT = percentage difference of the time through the movement between hip and knee flexion; %PkDORSI = percentage of the movement in which peak dorsiflexion occurs.

**Table 1 sports-13-00277-t001:** Joanna Briggs Institute critical appraisal for cross-sectional studies summary.

Study	Question								Total
	1	2	3	4	5	6	7	8	
Amasay & Suprak, 2010 [39]	Y	Y	Y	Y	Y	N	Y	Y	88%
Brochie et al., 2014 [11]	Y	Y	Y	Y	N	N	Y	Y	75%
Chiu et al., 2014 [12]	Y	Y	Y	Y	N	N	Y	Y	75%
Guess et al., 2020 [38]	N	Y	Y	Y	N	N	Y	Y	63%
Johnston et al., 2015 [40]	Y	Y	Y	Y	Y	Y	Y	Y	100%
Kipp et al., 2020 [49]	Y	Y	Y	Y	Y	Y	Y	Y	100%
Kryszkowski et al., 2022 [50]	N	Y	Y	Y	N	N	Y	Y	63%
Mikołajec et al., 2023 [51]	Y	Y	Y	Y	N	N	Y	Y	75%
Miller et al., 2023 [14]	N	Y	Y	Y	N	N	Y	Y	63%
Morris et al., 2022 [52]	Y	Y	Y	Y	N	N	Y	Y	75%
Pereira et al., 2018 [45]	Y	Y	Y	Y	N	N	N	Y	63%
Rauch et al., 2020 [41]	Y	Y	Y	Y	Y	N	Y	Y	75%
Shalfawi et al., 2014 [53]	Y	Y	U	Y	N	N	U	Y	50%
Shalfawi et al., 2011 [54]	Y	Y	Y	Y	N	N	Y	Y	75%
Shinkle et al., 2012 [55]	Y	Y	Y	Y	N	N	Y	Y	75%
Sole et al., 2018 [42]	Y	Y	Y	Y	N	N	Y	Y	75%
Soslu et al., 2016 [56]	Y	Y	Y	Y	N	N	Y	Y	75%
Swinton et al., 2014 [13]	Y	Y	Y	Y	Y	Y	Y	Y	100%
Average									76%

Y, yes; N, no; U, unsure.

**Table 2 sports-13-00277-t002:** Participant characteristics of included studies.

Study	*n*	Sex	Age (y)	Height (cm)	Weight (kg)	Groups
Amasay & Suprak, 2010 [39]	49	NS	20.2 ± 1.5	175.3 ± 8.6	73.8 ± 10.6	NCAA D2 Athletes
Brochie et al., 2014 [11]	16	M	26.7 ± 4	177 ± 4.3	72.1 ± 5.4	Qatar National Football Team
Chiu et al., 2014 [12]	16	F	20.4 ± 1.5	178 ± 6	70.1 ± 7.1	NCAA D1 Volleyball Players
Guess et al., 2020 [38]	394	NS	18-23	NS	74 ± 14.3	NCAA D1 Athletes
Johnston et al., 2015 [40]	50	M	21.7 ± 3.1	182 ± 8	77.4 ± 9.6	Competitive Basketball Players
Kipp et al., 2020 [49]	11	M	21.6 ± 1.8	193.3 ± 10.2	80.5 ± 10.5	NCAA D1 Basketball Players
Kryszkowski et al., 2022 [50]	22	M	20 ± 2	199 ± 6	93.8 ± 7.5	NCAA D1 Basketball Players
Mikołajec et al., 2023 [51]	15	NS	26.9 ± 8.3	191 ± 5.3	90.7 ± 10	Polish National 3 × 3 Basketball Team
Miller et al., 2023 [14]	2258	M and F	19.8 ± 1.3	178.5 ± 7.5	80.4 ± 15.8	NCAA D1 Athletes
Morris et al., 2022 [52]	14	M	22.7 ± 3.6	188 ± 8	88.2 ± 9.4	Competitive ARF Players
Pereira et al., 2018 [45]	15	M and F	28.3 ± 3.2	188.3 ± 4.6	93.5 ± 8.2	Brazilian National Handball Teams
Rauch et al., 2020 [41]	178	M	23.6 ± 3.7	200.3 ± 8	99.4 ± 11.7	NBA Players
Shalfawi et al., 2014 [53]	33	M	27.4 ± 3.3	192 ± 8.2	89.8 ± 11.1	Norwegian Pro Basketball Players
Shalfawi et al., 2011 [54]	30	F	19 ± 4	167 ± 4	57.5 ± 6.9	Competitive Norwegian Football Players
Shinkle et al., 2012 [55]	25	NS	19 ± 1.1	184 ± 4.6	106.2 ± 20.9	NCAA D1 American Football Players
Sole et al., 2018 [42]	150	M and F	20.3 ± 1.3	175.6 ± 7.3	75.1 ± 11.3	NCAA D1 Athletes
Soslu et al., 2016 [56]	23	M	23.2 ± 3.7	197.1 ± 9.1	95.3 ± 10.5	NS
Swinton et al., 2014 [13]	30	M	24.2 ± 3.9	182.4 ± 6.7	94.1 ± 12.3	Competitive Scottish Rugby Players

*n,* number of participants; NS, not specified; M, male; F, female; NCAA, National Association of Intercollegiate Athletics, D2, division two; D1, division one; ARF, Australian-rules football; NBA; National Basketball Association; Pro, professional.

**Table 3 sports-13-00277-t003:** Intervention characteristics of included studies.

Study	Equipment	Repetitions	Rest	Intensity	Cueing
Amasay & Suprak, 2010 [39]	Force Plates and Yardstick	3	120 s	100%	Height and Speed
Brochie et al., 2014 [11]	Force Plates	3	20 s	100%	Height
Chiu et al., 2014 [12]	Force Plates and Mo Cap	4	NS	100%	Height
Guess et al., 2020 [38]	Force Plates	2	NS	100%	Height and Speed
Johnston et al., 2015 [40]	Force Plates and Mo Cap	NS	NS	NS	NS
Kipp et al., 2020 [49]	Force Plates and Mo Cap	2-3 + 2 optional	NS	25–100%	NS
Kryszkowski et al., 2022 [50]	Force Plates	3	120 s	100%	Height and Speed
Mikołajec et al., 2023 [51]	OMD	3	NS	100%	Effort
Miller et al., 2023 [14]	Mo Cap	NS	NS	NS	NS
Morris et al., 2022 [52]	Force Plates	3	60 s	100%	Effort
Pereira et al., 2018 [45]	Jump Mat	5	15 s	100%	Effort
Rauch et al., 2020 [41]	Force Plates, Yardstick, and Mo Cap	3	90 s	100%	Effort
Shalfawi et al., 2014 [53]	Jump Mat	2	180 s	100%	Effort
Shalfawi et al., 2011 [54]	Force Plates	2	300 s	100%	Effort
Shinkle et al., 2012 [55]	Yardstick	1	180–300 s	100%	Effort
Sole et al., 2018 [42]	Force Plates	2	60 s	100%	Effort
Soslu et al., 2016 [56]	Force Plates	3	30 s	100%	Height
Swinton et al., 2014 [13]	Force Plates	2	NS	100%	Effort

Mo Cap, motion capture; OMD, optical measurement device; NS, not specified.

## Data Availability

The data that support the findings of this study are available upon request.

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
