# Peer review of "Physical and Biomechanical Relationships with Countermovement Jump Performance in Team Sports: Implications for Athletic Development and Injury Risk"

_sports, 2025, doi:10.3390/sports13080277_

Round 1

Reviewer 1 Report

Comments and Suggestions for Authors

The present systematic review sought to evaluate the characteristics associated with countermovement jump (CMJ) performance in adult team-sport athletes. A comprehensive search of three databases and grey literature yielded 18 articles that met the inclusion criteria. The strongest correlations included time-to-bottom, time-to-peak force, knee extension peak power at 180°/s, and squat jump height. The conclusions drawn from this study suggest that, in order to maximise CMJ performance, priority should be given to movement biomechanics and lower-body power, whilst taking into account individual braking-phase strategies. A few minor revisions are required to enhance the quality of the work:

In the Results section:
1) Please, improve the quality of Figure 1 and expand the caption.
2) Please, improve the readability of the labels on the x-axis in figures 4 and 5.
3) The conclusions section is short, so try to expand it by emphasising the purpose and impact of the work in the field of research.

Author Response

Comments 1: The present systematic review sought to evaluate the characteristics associated with countermovement jump (CMJ) performance in adult team-sport athletes. A comprehensive search of three databases and grey literature yielded 18 articles that met the inclusion criteria. The strongest correlations included time-to-bottom, time-to-peak force, knee extension peak power at 180°/s, and squat jump height. The conclusions drawn from this study suggest that, in order to maximise CMJ performance, priority should be given to movement biomechanics and lower-body power, whilst taking into account individual braking-phase strategies.

Response 1: Thank you for your suggestion. I have revised the appropriate sections of the abstract based on your wording and have highlighted this in red on page one, lines 12-15 and 19-23. 

Comments 2: Please, improve the quality of Figure 1 and expand the caption.

Response 2: Thank you for your suggestion. I have uploaded a higher quality version of figure 1 and expanded the caption accordingly. This can be found on page five, lines 166-167.

Comments 3: Please, improve the readability of the labels on the x-axis in figures 4 and 5.

Response 3: Thank you for your suggestion. I have enhanced the readability of the x-axis labels on figures 4 and 5 which can be found on pages nine and 10, lines 230 and 248 respectively.

Comments 4: The conclusions section is short, so try to expand it by emphasising the purpose and impact of the work in the field of research.

Response 4: Thank you for your suggestion. I have expanded upon the conclusion based on your pointed and have highlighted this in red on pages 16 and 17, lines 475-477, 486-494.

Reviewer 2 Report

Comments and Suggestions for Authors

Dear authors

Tranks for work in prepared in this manuscript. I have any recommendations that I think could better your work.

Abstract

It's a correct and accurate, only in the keyword could review if this keywords are in MESH or DECS

Introduction

This is the weakest section of the article. It is suggested that the authors delve deeper into several relevant topics, which they only cite and do not elaborate on.

  1. They should theoretically justify why they chose the classification of dimensions used to analyze the CMJ (physical characteristics, strength, power, joints in the CMJ, etc.). This is very relevant and is not included in the introduction.

  2. They should explain the phases of the CMJ and provide an overview, from which phase each variable originates (for example, in the results section, they talk about RFD; I imagine it is the eccentric phase, but they don't mention it anywhere).

  3. They mention the concept of "unclear." However, they do not explain the objective causes of this concept with the CMJ and its variables.

Materials and methods

This section is very well constructed. However, a more elaborate
graphic that perhaps combines the CMJ phases with each variable
would be advisable. It would be much more accessible to readers,
and would offer more effective communication.

Results

In table 2, it´s neccesary add more information about level, because the level that autor show are only national and not nomenclature international.

Its necessary add a table with results, for example: Authors, date, sample, instruments, variables, materials or technology, results.This is important for reader.

The results, while correct, are not compelling from a communication perspective (tables). They should offer a clearer sample of the results for the reader, as the analyses are not novel.

Discusión

This section is very well constructed. But the weak introduction hinders the discussion. The introduction needs to be significantly improved.

Author Response

Comments 1: Abstract - It's a correct and accurate, only in the keyword could review if this keywords are in MESH or DECS.

Response 1: Thank you for your suggestion. I have revised the keywords with their MESH format, where appropriate, and have highlighted this in red on page one, lines 26-27.

Comments 2: Introduction - They should theoretically justify why they chose the classification of dimensions used to analyze the CMJ (physical characteristics, strength, power, joints in the CMJ, etc.). This is very relevant and is not included in the introduction.

Response 2: Thank you for your suggestion. I have edited the introduction to include justification behind the selection of classifications for the different CMJ characteristics reported in the results section and have highlighted this in red on page two, line 72-76. 

Comments 3: Introduction - They should explain the phases of the CMJ and provide an overview, from which phase each variable originates (for example, in the results section, they talk about RFD; I imagine it is the eccentric phase, but they don't mention it anywhere).

Response 3: Thank you for your suggestion. I have edited the introduction to include a brief explanation of the relevant phases of the CMJ, while also supplementing and elaborating further with a figure in the materials and methods section. These have been highlighted in red on page two, line 55-57, and page four, line 145-149.

Comments 4: Introduction - They mention the concept of "unclear." However, they do not explain the objective causes of this concept with the CMJ and its variables.

Response 4: Thank you for your suggestion. I have amended the introduction to include an explanation behind the objectives causes for the lack of clarity around relative importance for each characteristic. This has been highlighted in red on page two, line 59-66.

Comments 5: Results - In table 2, it's neccesary add more information about level, because the level that autor show are only national and not nomenclature international.

Response 5: Thank you for your suggestion. I have edited table 2 with the suggestions made and have highlighted this in red on page six, line 192-193. 

Comments 5: Results - Its necessary add a table with results, for example: Authors, date, sample, instruments, variables, materials or technology, results.This is important for reader.

Response 5: Thank you for your suggestion. I have included a table with a summary of the equipment, repetitions, rest, intensity, and instructions delivered for each study included in the review. This has been highlighted in red on page seven, lines 206-207.

Round 2

Reviewer 1 Report

Comments and Suggestions for Authors

The Paper can continue the review process

Author Response

Comments 1: The Paper can continue the review process

Response 1: Thank you very much for your feedback!

Reviewer 2 Report

Comments and Suggestions for Authors

Dear authors

I would like to thank you for all the improvements you made to the work. However, some details are important changes for readers.

You mention "physiology" in the title. However, all the variables analyzed, found, and detailed are biomechanical and anthropometric. None are physiological; this should be addressed.

Please clarify these points.

Best regards.

Author Response

Comment 1: You mention "physiology" in the title. However, all the variables analyzed, found, and detailed are biomechanical and anthropometric. None are physiological; this should be addressed. Please clarify these points.

Response 1: Thank you for your follow up feedback. We understand the point you have raised and we have made the appropriate changes. This includes replacing the specific mentioning of the term "physiological" and related synonyms and replacing these with "physical". Similarly, to avoid a clash of terminology with the original "physical characteristics" results category, this was relabelled as "anthropometric and morphological characteristics", as related to your suggestions. These changes can be found throughout the document, including pages 1 (line 2, 10, 37, and 39), 2 (line 75), 3 (line 129-130, 133-135), 7 (line 211-212 and 216), 8 (line 222-223), 13 (line 333, 335 and 337), 14 (line 378), and 17 (line 498). We hope these changes provide a sufficient connection between the study conclusions and results.